# Interactions of Ingested Polystyrene Microplastics with Heavy Metals (Cadmium or Silver) as Environmental Pollutants: A Comprehensive In Vivo Study Using *Drosophila melanogaster*

**DOI:** 10.3390/biology11101470

**Published:** 2022-10-08

**Authors:** Fatma Turna Demir, Gökhan Akkoyunlu, Eşref Demir

**Affiliations:** 1Department of Medical Services and Techniques, Medical Laboratory Techniques Programme, Vocational School of Health Services, Antalya Bilim University, 07190 Antalya, Turkey; 2Department of Histology and Embryology, Faculty of Medicine, Akdeniz University, 07070 Antalya, Turkey

**Keywords:** *Drosophila melanogaster*, polystyrene microplastics, heavy metals, cadmium, silver, uptake, Comet assay, oxidative stress, gut damage, carriers

## Abstract

**Simple Summary:**

Living organisms are now constantly exposed to PSMPLs, and besides their huge toxic potential, they can also act as carriers of various hazardous elements such as heavy metals. As a novelty, we managed to visualize the biodistribution of ingested PSMPLs throughout the fly’s body, observing the interactions of such plastics with *Drosophila* intestinal lumen, and cellular uptake by hemocytes. This study is the first investigation to investigate the various biological effects of the interactions between ingested PSMPLs and heavy metals in *Drosophila*.

**Abstract:**

Living organisms are now constantly exposed to microplastics and nanoplastics (MNPLs), and besides their toxic potential, they can also act as carriers of various hazardous elements such as heavy metals. Therefore, this study explored possible interactions between polystyrene microplastics (PSMPLs) and two metal pollutants: cadmium chloride (CdCl_2_) and silver nitrate (AgNO_3_). To better understand the extent of biological effects caused by different sizes of PSMPLs, we conducted in vivo experiments with five doses (from 0.01 to 10 mM) that contained polystyrene particles measuring 4, 10, and 20 µm in size on *Drosophila* larvae. Additional experiments were performed by exposing larvae to two individual metals, CdCl_2_ (0.5 mM) and AgNO_3_ (0.5 mM), as well as combined exposure to PSMPLs (0.01 and 10 mM) and these metals, in an attempt to gain new insight into health risks of such co-exposure. Using transmission electron microscopy imaging, we managed to visualize the biodistribution of ingested PSMPLs throughout the fly’s body, observing the interactions of such plastics with *Drosophila* intestinal lumen, cellular uptake by gut enterocytes, the passage of plastic particles through the intestinal barrier to leak into the hemolymph, and cellular uptake by hemocytes. Observations detected size and shape changes in the ingested PSMPLs. Egg-to-adult viability screening revealed no significant toxicity upon exposure to individual doses of tested materials; however, the combined exposure to plastic and metal particles induced aggravated genotoxic effects, including intestinal damage, genetic damage, and intracellular oxidative stress (ROS generation), with smaller sized plastic particles + metals (cadmium and silver) causing greater damage.

## 1. Introduction

The amount of improperly discarded plastic waste accumulating in the world’s waters, especially in subtropical gyres, has reached alarming levels [1]. Although there have been hasty campaigns to clean up oceans, global plastic debris is expected to triple in volume unless there are sustained improvements in the management of plastic waste [2]. Historically, much of the environmental concern has been geared towards the consequences of bulk plastic waste, but recent research is now investigating the health hazards of micro and nano plastic particles that form as a result of the degradation of such bulk waste [3,4]. Such particles, known as secondary microplastics and nanoplastics (MNPLs), could be even more harmful contaminants, so their potential impact on life on Earth should be thoroughly analyzed. Plastic has been used in many consumer and industrial products such as toys, food packaging, automobiles, and electronics [4,5,6,7,8], as well as in cosmetic and personal care products [1,2,3,4,5,6,7,8,9].

Exposure to MNPLs in humans can occur via several routes, including skin contact with personal care products such as creams [9]; ingestion, primarily of seafood [10,11] and in other foods [4,12,13,14]; and inhalation [15,16,17]. Bouwmeester et al. [6] reported that beverages such as beers and lagers contained similar amounts of MNPLs to those in certain seafoods such as mussels. Such a prevalent presence calls for an urgent and detailed risk assessment to determine their health hazards to human life and all other life forms [18,19]. Some ecotoxicity research has reported certain biological effects of MNPLs on in vivo models [11,17,20,21,22]; however, we still have insufficient data on the toxic profile of these particles, particularly the harms of ingested MNPLs in mammals. A range of industries utilize metals and metalloids as dyes and paints, as well as UV and heat stabilizers in plastic products that are designed as food or drink containers or packages [23,24,25]. Silver (Ag) content in the surface soils varies between <0.01 and 5 mg/kg [26,27]. In addition, the general concentration of cadmium (Cd) ranges from 0.1 to 1 mg/kg in soil [28]. In this context, Cd and its variants are widely employed in plastics from which they can easily dissolve into foods and beverages [29,30,31,32]. Another common metal in consumer products such as deodorants, clothing materials, bandages, and in cleaning solutions [33,34], Ag is mainly preferred for its excellent antibacterial and antifungal properties. Thanks to their greater surface areas and polarity, MPLs can represent a crucial vehicle for the carrier of heavy metals in the marine environment [35], so they can act as carriers for heavy metals [36]. On that note, MPLs may show variations in their route of exposure and dose, as they can transport adsorbed metals into organisms, resulting in metal accumulation in the digestive system [37]. Nevertheless, the health risks associated with MPLs acting as carriers is still a hotly debated issue. As known, MNPLs widely coexist with other pollutants in the environment at relatively low concentrations [38]. Research indicates that MPLs can alter the localization of metals in living tissues and cells [37]. Heavy metal particles transported by ingested MPLs within tissues may cause greater exposure and downstream effects by means of additive effects or synergistic activities [39,40]. In the meantime, low-density MPLs such as polyethylene (PET) MPLs have been claimed to decrease metal bioavailability in oceans and inland waters, for they often float on the surface, and thus, minimize the chance of interaction between aquatic organisms and adsorbed metals [37,41]. Such contradictory findings and conclusions indicate that we need better insight into the combined activities of MPLs and metals.

The accurate quantitative measurement of toxicity and potential risks associated with MNPL exposure via various routes requires strenuous scientific effort involving in vivo experiments using mammalian organisms such as rodents [4,42]. For example, experiments on mice suggest that MNPL accumulation could induce deleterious effects, but its impact over the long term has yet to be elucidated [43]. *Drosophila melanogaster* (fruit fly) has been used in various studies aiming to show the adverse biological effects (morphological changes, gut damage, phenotypic and behavioral effect, fertility, oxidative stress, gene expression and epigenetic effects, metabolic diseases, etc.) of different types of micro/nanoplastics [7,44,45,46,47,48,49,50,51]. Along with its ability to produce many generations in a short time (over 20 generations each year), *D. melanogaster*, on the other hand, allows scientists to circumvent common ethical restrictions placed on the use of higher vertebrates in experiments, namely, the principles of the three Rs: replacement, refinement, and reduction, which govern the treatment of laboratory animals [52]. In view of such complications, this fruit fly species stands out as the most convenient model among all alternative testing models [7,53,54]. Moreover, pressure from the public and animal rights institutions has forced researchers to conduct in vitro studies instead of experimenting on mammals [8,55,56]. Therefore, they have recently resorted to using lower eukaryotic models such as fruit flies (*D. melanogaster)* [7,57,58], zebrafish (*Danio rerio*) [59,60], and roundworms (*Caenorhabditis elegans*) [61]. Numerous studies have begun to carry out nanotoxicity and nanogenotoxicity experiments on *D. melanogaster*, using somatic cells as targets to assess harmful effects on exposed flies without waiting for subsequent generations. As a result of such technical advantages, *Drosophila* is now regarded as a reliable and dynamic tool in measuring the potential impact of plastic contaminants. In addition, the European Center for the Validation of Alternative Methods (ECVAM) recommends the use of *D. melanogaster* in toxicological research and environmental monitoring activities [62,63]. The prevalent use of *D. melanogaster* in toxicity research has led to the establishment of a novel field: Drosophotoxicology [64]. This relatively new term refers to a plethora of methods where researchers utilize *D. melanogaster* as an experimental model organism [65]. The DNA of the fruit fly contains genes that are homologous (at around 75% similarity) to those found in humans and known to be responsible for diseases such as gastrointestinal infections [57], cardiovascular diseases [66], cancer [67], spinocerebellar ataxia [68], and neurodegenerative diseases including Parkinson’s and Alzheimer’s [69,70]. Furthermore, *Drosophila* flies display relatively complex motor behaviors such as walking, climbing, and flying, so despite their much simpler nervous system, they are employed in cognitive and memory research involving experiments with conditioned fear responses [71]. Humans and fruit flies also share several biological commonalities, including cell proliferation, gene expression, cell signaling, innate immune signaling, cell differentiation, cellular homeostasis, and apoptosis [72,73,74,75].

Although MNPLs are thought to pose certain health risks both in their own right and as carriers of other contaminants, we still lack evidence for their combined effects. The number of studies employing *Drosophila* in their experiments to determine the potential hazards of PSMPLs exposure has been limited. Therefore, this study aimed to assess the interactions between PSMPLs and heavy metals such as cadmium chloride (CdCl_2_) or silver nitrate (AgNO_3_), and whether their combined exposure caused toxic and genotoxic impacts on *Drosophila* larvae. Lethality, intestinal damage, the translocation of ingested PSMPLs using transmission electron microscopy (TEM), intracellular ROS production in hemocytes, and DNA damage in hemocytes were tested with a Comet assay.

## 2. Materials and Methods

### 2.1. Chemicals

Micro particle size standards based on polystyrene monodisperse analytical standard (CAS No. 9003-53-6, sizes: 4, 10, and 20 µm) were purchased from Sigma Chemical Co. (St. Louis, MO, USA). Coulter Multisizer III (Worcestershire, UK) was used for detecting the size analyses of the PSMPLs according to the Sigma-Aldrich certificate of analysis. The PSMPLs were determined as particle specific gravity 1.05 g/cm^3^ at 20 °C, solid content: 2% water and polystyrene for particle composition. Cadmium chloride (CdCl_2_, 99% purity; CAS No. 10108-64-2) was purchased from Acros Organics (Geel, Belgium) and silver nitrate (AgNO_3_, 99.0% purity; CAS No. 7761-88-8) from Carlo Erba (Val de Reuil, France). Prior to use, PSMPLs, CdCl_2_, and AgNO_3_ were dispersed in sterile distilled water, which served as a solvent control, while 4 mM of ethyl methanesulfonate (EMS) was used as a positive control for the Comet assay (SCGE), and 0.5 mM of hydrogen peroxide (H_2_O_2_) for the assessment of reactive oxygen species (ROS).

### 2.2. D. melanogaster Strains, Exposure, and Viability (Lethality)

*Drosophila* larvae and adult flies were cultured at room temperature (25 °C) and 60% humidity through a food medium containing sugar, cornmeal, yeast, agar, propionic acid, and nipagin. The wild-type strain Oregon-R^+^ flies were employed in all the experiments. The procedure followed in the experiments to detect the toxicity and genotoxicity of PSMPLs, CdCl_2_, and AgNO_3_ was as follows: (i) In the first group, three-day-old (72±4 h) *Drosophila* larvae were exposed to PSMPLs (0.01, 0.1, 0.5, 1, 2.5, 5, and 10 mM); CdCl_2_ (0.01, 0.1, 0.5, 1, 2.5, 5, and 10 mM); and AgNO_3_ (0.01, 0.1, 0.5, 1, 2.5, 5, and 10 mM). (ii) In the second group, 72±4 h-old *Drosophila* larvae were exposed to both concentrations of PSMPLs (0.01 and 10 mM) and CdCl_2_ (0.5 mM) simultaneously. (iii) In the third group, 72±4 h *Drosophila* larvae were exposed to the concentrations of PSMPLs (0.01 and 10 mM) and AgNO_3_ (0.5 mM) simultaneously.

The impact of exposure to three sizes of PSMPLs (4, 10, or 20 µm) was measured by observing egg-to-adult survival rates. Adult flies were placed in bottles containing food medium, which were darkened by carbon powder application to allow easy egg collection every 8 h. The experimental samples, each consisting of 50 fly eggs, were then transferred to plastic vials containing 4 g of *Drosophila* food culture (Carolina Biological Supply Co. Burlington, NC, USA), which was previously saturated with various concentrations (10 mL) of PSMPLs (0, 0.01, 0.1, 0.5, 1, 2.5, 5, and 10 mM). The doses corresponded to nominal concentrations of 1.0415, 10.415, 52.075, 104.15, 260.375, 520.75, and 1041.5 µg/mL, and once dispersed in the food culture represented 0.00260375, 0.0260375, 0.1301875, 0.260375, 0.6509375, 1.301875, and 2.60375 mg/g of food. The food culture was previously saturated with various concentrations (10 mL) of CdCl_2_ (0, 0.01, 0.1, 0.5, 1, 2.5, 5, and 10 mM). The doses corresponded to nominal concentrations of 1.8332, 18.332, 91.66, 183.32, 458.3, 916.6, and 1833.2 µg/mL, and once dispersed in the food culture represented 0.004583, 0.04583, 0.22915, 0.4583, 1.14575, 2.2915, and 4.583 mg/g of food. The food culture was saturated with various concentrations (10 mL) of AgNO_3_ (0, 0.01, 0.1, 0.5, 1, 2.5, 5, and 10 mM). These doses corresponded to nominal concentrations of 1.6987, 16.987, 84.935, 169.87, 424.675, 849.35, and 1698.7 µg/mL, and once dispersed in the food culture represented 0.00424675, 0.0424675, 0.2123375, 0.424675, 1.0616875, 2.123375, and 4.24675 mg/g of food. The concentrations of PSMPLs were based on our previous toxicity studies into particulate matter on *Drosophila* (0.01–10 mM) [44,45,47]. On the other hand, the concentrations of CdCl_2_ and AgNO_3_ were determined based on the data from previous toxicity and genotoxicity research with CdCl_2_ (0.01–0.2 mM) and AgNO_3_ (0.05, 0.2, and 1 mM) on *Drosophila* [33,44,50]. The highest PSMPL dose did not exceed 10 mM, and a total of five replicates (50 eggs for each replicate) were used per concentration. Following exposure to the test materials, the number of surviving adult flies was recorded to calculate the survival rate as compared to the controls. The PSMPLs and metal doses are provided in mM throughout the paper to avoid confusion and to maintain uniformity.

### 2.3. Trypan Blue Staining

Cell membrane impermeable trypan blue is one of several stains widely utilized to test cell survival. With minor modification, trypan blue staining, which is useful for the determination of gut effects in *Drosophila*, was conducted on the larvae based on a protocol previously reported in studies [44,76]. Four replicates (20 larvae for each replicate) were performed for each group. All larvae were washed with PBS (1×) solution and then were placed on 0.8% agarose (containing 5% sugar and 10% trypan blue stain) plates for 30 min. Following this procedure, they were washed with PBS again for 15 min, and in the final stage, they were analyzed through stereomicroscopy (SLX-2 STEREOZOOM, Ponteranica, Italy) to calculate the ratio of flies whose gut was stained blue as a sign of gut damage.

### 2.4. Interaction between PSMPLs and Intestinal Barrier Components

We followed the experimental protocol we described in our previous work to examine the biobehavior of PSMPLs (4, 10, and 20 µm) in various components of intestine [49,77,78,79,80]. After 5 days of exposure, a total of 10 larvae (2.60375 mg/g of food) were dissected in phosphate buffer solution to extract the intestinal tracts. The intestines were immersed in a fixation solution (4% glutaraldehyde prepared in 0.1 M of Sorensen phosphate buffer) for 2 h, post-fixed in 1% (*w*/*v*) osmium tetroxide containing 0.8% (*w*/*v*) potassium hexacyanoferrate, before washing four times with deionized water and sequential dehydration in acetone. After that, the samples were embedded in Eponate 12^™^ resin (Ted Pella Inc., Redding, CA, USA) and left to polymerize at 60 °C for 48 h. Sections of ultrathin (100 nm) were cut from the resin blocks by means of a diamond knife (45°, Diatome, Biel, Switzerland). They were then placed on non-coated 200 mesh copper grids and contrasted with uranyl acetate for 30 min and Reynolds lead citrate for 5 min. The imaging of these sections was performed with transmission electron microscopy (Zeiss Leo 906E, Austin, TX, USA) equipped with a CCD camera (Erlangshen ES1000W, Gatan Inc., Pleasanton, CA, USA). During this stage of the experiments, we only used the higher doses of MPLs for easier imaging of their biodistribution through the intestinal tract of *Drosophila*.

### 2.5. Detection of PSMPLs in the Hemolymph

The hemolymph samples from *Drosophila* larvae were examined to check any potential translocation of PSMPLs through the intestinal barrier. Accordingly, after 5 days of exposure, the samples were prepared and investigated in accordance with the protocol we previously described in detail [80]. In short, the larvae were collected with forceps, cleaned, and dissected in buffer solution (PBS 1%) to obtain the hemolymph fluid. The hemolymph samples were placed on copper grids, and once they were dry, we checked whether they contained PSMPLs using TEM imaging (Zeiss Leo 906E, Austin, TX, USA).

### 2.6. Intracellular Oxidative Stress (Reactive Oxygen Species, ROS) Detection

We evaluated the intracellular levels of ROS in hemocytes of *Drosophila* larvae after exposure to PSMPLs, CdCl_2_, AgNO_3_, as well as combined exposure to these metals and PSMPLs. In accordance with our previous protocol [57,77,78,79,80,81,82], we collected hemocyte samples and incubated them with 5 μM of DCFH-DA for 30 min at 24 °C. The effects were examined through a fluorescence microscope with an excitation of 485 nm and an emission of 530 nm (green filter). The positive control was hydrogen peroxide (H_2_O_2_, 0.5 mM). Images obtained by fluorescence microscopy were analyzed on the ImageJ software package.

### 2.7. Genotoxicity Induction (Comet Assay)

Commonly utilized in genotoxicity investigations for the prediction of cancer, the Comet assay is a highly versatile method to detect DNA strand breaks in the hemocytes of *Drosophila* larvae. Hemocytes were isolated in accordance with the protocol described by Irving et al. [83]. With minor modifications, the Comet assay protocol previously laid out by Singh et al. [84] was employed for the isolated hemocyte samples, which were investigated for cell viability under light microscope with trypan blue stain [85]. The slides for the microscope were coated with 1% normal melting point (NMP) and then dried. For each experimental group, 20 µL of isolated hemocytes (approximately 40,000 cells) were mixed with 120 µL of 75% low melting point agarose (LMA). LMA was dissolved in phosphate-buffered saline (PBS). Hemocytes mixed with LMA were placed onto NMP-coated slides to which cover slips were added, and kept on ice for 5 min. The coverslips were removed, and 80 µL of LMA was added onto the slides and then kept on ice for 5 min once again [84]. Next, the coverslips were removed, and the slides were soaked in cold, freshly-prepared lysis solution (10 mM of Trisma base, 100 mM of EDTA, 2.5 M of NaCl, 1% Triton X-100 and 1% N lauroylsarcosinate pH 10) for 2 h at 4 °C in a dark chamber. Dimethyl sulfoxide was not used in the lysing solution, as it causes damage in the tissues of *Drosophila* [86].

To prevent additional DNA damage that could be caused by light, the procedure was carried out in the dark. The slides were initially kept in a cold buffer (1 mM of EDTA and 300 mM of NaOH, pH 12.8) for 20 min and electrophoresed for 20 min (300 mA and 1 V/cm) at 4 °C. After electrophoresis, the slides were washed with neutralization buffer (Tris buffer, pH 7.5) and stained with ethidium bromide (20 mg/mL). Finally, they were washed with distilled water, photographed with a fluorescence microscope (filter 515–560 nm), and Comet image analyses were performed on CASP Lab software (version 1.2.3b2) [87], as described in previous studies [88,89]. CASP Lab software is a reliable software utilized to detect DNA damage in comet images [87,88,89,90]. Triplicates of 100 randomly selected cells were analyzed per treatment (total of 300 cells). The percentage of DNA in the tail (% DNA tail) was used as a measure of DNA damage. Values for mean and standard error were calculated in terms of % DNA tail.

### 2.8. Statistical Analysis

Unless stated otherwise, the research data are presented as means of two independent experiments, including duplicates of each one. Arithmetic values for mean ± standard error were calculated for the assays. Prior to the statistical analysis, the normality tests (Kolmogorov–Smirnov and Shapiro–Wilk tests) were carried out on SigmaPlot version 11.0 (SPSS, Chicago, IL, USA) and equal variances across samples were determined by Levene’s test. Data following normal distribution and equal variance (Comet assay) were further analyzed by a Student’s *t*-test on SigmaPlot version 11.0. A Mann–Whitney *U*-test was used to analyze toxicity/lethality, gut damage, and ROS production. Statistical significance was accepted as *p* < 0.05.

## 3. Results

The negative control (sterile distilled water) was first checked for toxicity, and *Drosophila* eggs that were exposed to this negative control showed 100% viability. Viability experiments were conducted in our laboratory to determine the range of concentrations to be used in the experiments. The criteria to choose the final selected concentrations were chosen for two reasons: (1) a decrease in the percentage of developing exposed larvae is a clear indication that the chemicals affected the larvae; (2) the number of emerging adults should be high enough to perform the experiments. Instead, we performed a toxicity study to detect the genotoxic effects at concentrations above 50 percent viability. According to these results, 0.01, 0.1, 0.5, 1, 2.5 5, and 10 mM concentrations of PSMPLs (4, 10, and 20 µm) have no statistically significant toxic effects as in *D. melanogaster* larvae. Significant toxic effects were observed after exposure to CdCl_2_ and AgNO_3_ at doses higher than 0.5 mM (i.e., 1, 2.5, 5, and 10 mM). Thus, we have determined 0.01, 0.1, and 0.5 mM as non-toxic concentrations for CdCl_2_ and AgNO_3._

### 3.1. Gut Damage after Exposure to PSMPLs, CdCl2, AgNO3, and Combined Exposure

We used trypan blue staining to detect intestinal cell damage after exposure to PSMPLs, CdCl_2_, AgNO_3_, as well as combined exposure to PSMPLs + CdCl_2_ and PSMPLs + AgNO_3_. Positive trypan blue staining was only observed in exposure groups, including those exposed to PSMPLs alone, CdCl_2_ alone, AgNO_3_ alone, and those exposed to a combination of PSMPLs and CdCl_2_ or AgNO_3_ (Figure 1A), which showed that such materials caused intestinal damage. When compared with the distilled water, exposure to PSMPLs (4, 10, and 20 µm) alone induced greater intestinal damage at 36% (*p* ≤ 0.001), 32% (*p* ≤ 0.001), and 30% (*p* ≤ 0.001) at the highest concentration (10 mM) (Figure 1B). Moreover, exposure to CdCl_2_ and AgNO_3_ caused intestinal damage to a certain extent. The highest dose (0.5 mM) of CdCl_2_ and AgNO_3_ increased the intestinal damage by 69% (*p* ≤ 0.001) and 61% (*p* ≤ 0.001), as compared to the negative control (Figure 1C). As for the combined exposure to plastics and metals, the highest dose (10 mM) of PSMPLs (4, 10, and 20 µm) + CdCl_2_ (0.5 mM) increased the intestinal damage by 83% (*p* ≤ 0.001), 75% (*p* ≤ 0.001), and 70% (*p* ≤ 0.001), as compared to exposure to CdCl_2_ alone (Figure 1D). On the other hand, the highest dose (10 mM) of PSMPLs (4, 10, and 20 µm) + AgNO_3_ (0.5 mM) caused relatively less gut damage at 78% (*p* ≤ 0.001), 71% (*p* ≤ 0.001), and 67% (*p* ≤ 0.001) when compared with exposure to AgNO_3_ alone (Figure 1E). PSMPLs + CdCl_2_ co-treatment is higher than PSMPLs + AgNO_3_ co-treatment. High staining, as shown in the larvae, supposes a high amount of gut damage, which must produce more toxicity than observed.

### 3.2. Monitoring of PSMPLs along the Intestinal Tract

The biobehavior of PSMPLs during their biodistribution through the intestinal tract provides valuable data on their potential biological impact. Therefore, we investigated this biodistribution step by step from their entrance into the intestinal lumen to their interaction with various lumen components, the intestinal barrier to their cellular uptake and distribution within enterocytes, and their possible translocation into the hemolymph. Figure 2 and Figure 3 illustrate these processes. The imaging results of the smallest particles (4 µm) are given in Figure 2A–D. The MPL particles were attached to the peritrophic membrane (Figure 2A), the first line of the intestinal barrier defense, and they were detected inside the microvilli (Figure 2B) and distributed in the midgut lumen (Figure 2C). PSMPLs (4 µm) translocated through the intestinal barrier (Figure 2D). The imaging results of PSMPLs (10 and 20 µm) from ingestion to their translocation into the circulatory system are given in Figure 3A–C. Upon ingestion of 10 and 20 µm MPLs, their existence inside the midgut lumen can be seen in Figure 3A,B. Internalized by the intestinal barrier, PSMPLs measuring 20 µm found their way inside the microvilli (Figure 3B). The ingested PSMPLs (10 and 20 µm) ultimately ended up in the hemolymph (Figure 3C). The findings demonstrated that such plastic particles, regardless of their size, could penetrate and cross the intestinal barrier to reach the open circulatory system.

### 3.3. Oxidative Stress Induction

Oxidative stress status is an important factor in demonstrating the adverse effects of individual exposure to PSMPLs, CdCl_2_, AgNO_3_, and combined exposure to these plastics and metals. In Figure 4A–D, ROS production in the hemocytes of third-instar larvae can be seen after exposure to study compounds, along with their comparison to controls. The potential presence of PSMPLs in the hemolymph indicates that these materials might interact with hemocytes, causing substantial biological effects. In our study, exposure to PSMPLs (4, 10, and 20 µm) elevated ROS production in *Drosophila* hemocytes depending on size and dose, reaching significant levels at the highest doses (5 and 10 mM). The 10 mM dose of PSMPLs (4 µm) was found to cause the greatest oxidative stress. ROS generation was much higher after exposure to smaller particles (4 µm) (301%), as compared to larger ones (10 and 20 µm) (287 and 275%, respectively), suggesting that PSMPLs (4 µm) caused greater ROS production (Figure 4A). CdCl_2_ and AgNO_3_ also caused a certain degree of ROS production in hemocytes. The highest concentration (0.5 mM) of CdCl_2_ and AgNO_3_ increased the ROS production rate by 287 and 272% compared with the control (Figure 4B). For the combined exposure, the highest concentration (10 mM) of PSMPLs (4, 10, and 20 µm) + CdCl_2_ (0.5 mM), respectively, increased the ROS production rate by 339, 318, and 306% compared with the CdCl_2_-alone group (Figure 4C). In Figure 4D, the highest concentration (10 mM) of PSMPLs (4, 10, and 20 µm) + AgNO_3_ (0.5 mM), respectively, increased the ROS production rate by 326, 311, and 301% compared with the AgNO_3_-alone group. The ROS values were higher in PSMPLs (4 µm) + CdCl_2_ (0.5 mM) (339%) than in PSMPLs (10 and 20 µm) + CdCl_2_ (0.5 mM) or PSMPLs (10 and 20 µm) + AgNO_3_ (0.5 mM) (Figure 4C).

### 3.4. Genotoxic Effects Determined by Comet Assay

Exposure to the different sized PSMPLs (4, 10, and 20 µm) induced significant increases in the levels of DNA damage at the highest tested concentrations (5 and 10 mM) (Figure 5A). Despite the genotoxicity, to a certain extent, of all sizes of PSMPLs, exposure to smaller particles (4 µm) induced greater DNA damage. The highest concentration (0.5 mM) of CdCl_2_ and AgNO_3_ increased the DNA damage in hemocytes (Figure 5B). As for the combined exposure, the highest dose (10 mM) of PSMPLs (4, 10, and 20 µm) + CdCl_2_ (0.5 mM), respectively, increased the DNA damage rate by 41.17, 38.61, and 36.82%, as compared with exposure to the negative control (Figure 5C). On the other hand, the highest concentration (10 mM) of PSMPLs (4, 10, and 20 µm) + AgNO_3_ (0.5 mM), respectively, increased the DNA damage rate by 39.85, 36.48, and 33.92% compared with the negative control (Figure 5D). The DNA damage values were higher in PSMPLs (4 µm) + CdCl_2_ (0.5 mM) (41.17% of DNA tail) than the others (Figure 5C). These findings indicate that concentration could be a primary determinant of harmful impacts.

## 4. Discussion

The current literature is scarce about the potential effects of exposure to MNPLs on *Drosophila* as a model organism. In this study, we therefore explored interactions between MPLs and heavy metals such as CdCl_2_ and AgNO_3_, and whether their combined exposure caused toxic and genotoxic impacts on *Drosophila* larvae, finding that exposure to such materials significantly impaired gut health and caused oxidative stress and DNA damage. After ingestion, the translocation of microparticles and/or PSMNPLs in the gut has been shown in animal models such as mussels [91], fish [92], rats [93], and insects [94]. Trypan blue staining is a common method to detect cell death, where the negatively charged dye only stains the compromised cell membrane [95]. Exposure to PSMNPLs was reported to cause intestinal damage, as revealed by differential staining of the damaged and unharmed intestines of exposed and unexposed larvae [44]. Likewise, our findings indicated that exposure to PSMPLs induced significant gut damage in *Drosophila* larvae. For some, intestinal damage could be associated with scratch or friction caused by MPs in the digestive tract [96], while others argue that MPs might induce elevated ROS production and trigger oxidative stress, thus, resulting in inflammation in the gut tissues [97]. Recent research involving in vivo experiments on *Drosophila* flies has found that exposure to PSMNPLs could have serious consequences, including an impaired gastrointestinal (GI) tract, locomotor dysfunction, aggravated cadmium toxicity, and epigenetic gene silencing [44], along with morphological defects, impaired climbing behavior, and somatic DNA recombination [45].

PETMNPL exposure has been reported to deteriorate fertility and result in a significantly smaller size of offspring [46], reduced oviposition in female flies, and lower triglyceride and glucose content in male flies [48]. Chronic exposure to PSMNPLs (1 μm and 20 nm) has recently been studied on *Drosophila* larvae by Matthews et al. [47], who observed markedly impaired locomotor behavior, without any significant changes in survival, reproduction capacity, and egg-to-adult development in flies [47]. More recent research has detected no toxic effects upon the ingestion of PSNPLs, yet they caused significant molecular responses leading to changes in the expression of certain genes, as well as ROS generation and DNA damage, with smaller particles causing stronger effects in *Drosophila* larvae [49]. Recent studies exploring potential interactions between Ag compounds (Ag nanoparticles, and AgNO_3_) and two different sizes of polystyrene nanoplastics (PSNPLs) (PS-50 and PS-500 nm) using *Drosophila* larvae support the finding of an antagonistic interaction between silver compounds and PSNPLs [50]. Another study investigated the impact of exposure to PETMPLs (2 µm) on *Drosophila*’s lifespan, reporting an increase in the lifespan of male flies after treatment with 1 g/L with no difference in female flies [51]. Overall, our findings appear to confirm the genotoxic potential of ingested PSMPLs, as smaller MPLs translocated via the intestinal barrier and reached the hemolymph in amounts that were large enough to cause significant damage in DNA. Nevertheless, no size range of the tested MPLs had any significant reduction in the viability or egg-to-adult survival of *Drosophila*, and such a lack of toxicity is in concordance with two previous studies using *Drosophila* to test the effects of PSMNPL in different size ranges [44,45,49]. This was an expected outcome, as PSNPLs were found to cause no toxic effects on different human Caco-2 cell lines, a commonly used in vitro model [13]. As ingestion is the primary route of exposure for MNPLs [98], comprehensive research is needed to better understand what happens during their biodistribution into the intestinal tract [99]. The gastrointestinal system of *Drosophila* has been regarded as a dynamic and versatile model in research on intestinal epithelial homeostasis and relevant changes caused by environmental contaminants [100]. From this perspective, we decided to apply the experience from our previous research on the internalization of nanomaterials in *Drosophila* gut to the case of MPLs [80]. Detailed TEM imaging studies enabled visualization of MNPLs engulfed by luminal membranes, probably stemming from the peritrophic matrix, which could be suggested as a potential mechanism for the intestinal cell uptake. The peritrophic matrix in *Drosophila’s* midgut is often thought to be the first defense line, with functions comparable to those of the mucus layer in humans, which is responsible for protecting the intestinal epithelial cells. The uptake of MNPLs by the intestinal microbiota has the potential to affect its functionality, so their role in the imbalance of gut microbiota should be confirmed by further research [101]. The fruit fly has been considered as a reliable and simple model organism in investigating the effects of environmental pollutants on the microbiome [102], since disturbances in its microbiome may lead to disease or the disruption of homeostasis [103]. When ingested, PSNPLs may interact with midgut lumen and various intestinal enzymes, together with the acidic environment in the intestines, potentially shrinking in size [49]. Such effects were confirmed in our study via measurements of particle diameters located in the midgut lumen and comparisons with those prior to ingestion. In fact, Yang et al. [104] reported that plastic-eating mealworms can biodegrade polystyrene in their gut by forming pits and cavities on its surface through digestive enzymatic activities. Another recent work has reported that intestinal digestive enzymes could cause the biodegradation of ingested TiO_2_ nanoparticles in *Drosophila* [102]. However, such biological effects, that is, the degradation ability of MNPLs by intestinal microbiota, have yet to be fully elucidated. Despite some evidence that members of the phyla Actinobacteria and Firmicutes from *Lumbricus terrestris* can reduce the size of MPLs within the gastrointestinal tract [105], we need future research to confirm whether this occurs in humans or other mammals [106].

Our in vivo model allowed easy visualization of the cellular uptake of MNPLs, their distribution inside the enterocytes that line the intestinal walls. This is a finding that has never been reported in studies using in vivo organisms such as *C. elegans* and zebrafish [107]. We discovered that MNPLs were distributed throughout the insect’s midgut microvilli and were bound to the peritrophic membrane, which is comparable to what was observed in the Caco-2 cell line, and such an interaction was reported to cause certain morphological alterations in the mitochondria in human cell cultures [13], with membrane disruption contributing to mitochondrial dysfunction and leading to a wide range of human pathologies [108]. Recent research reports observing the presence of MNPLs in the hemolymph in bivalve animals such as *Mitillus* [109], which further confirms their ability to cross intestinal barriers. MPLs can reach the different tissues and organs once in the hemolymph. Indeed, some studies detected that MNPLs could be incorporated into hemocytes as a result of their phagocytic activity. For instance, results similar to what we observed in *Drosophila* have also been reported in the hemocytes of the silkworm larvae [110] and *Mitillus* [109]. The cellular uptake of MNPLs by hemocytes has been reported in different organisms. However, they were ex vivo studies. For example, the hemocytes of a freshwater species called zebra mussels were exposed to different doses of MNPLs to explore their phagocytic activity [111], while some research applied this approach in *Drosophila* [112]. All of these studies seem to confirm the ability of MNPLs to cross the intestinal barrier and internalization by hemocytes, as well as cells of various other tissues. The intestinal uptake of MNPLs and their translocation into the hemolymph compartments may exert toxic effects on cells and organs, if not on the entire organism. The toxicity of ingested MNPLs (100 nm), depending on dose, have been reported to be mediated by ROS generation in *Daphnia pulex* [113], *Daphnia* [114], *C. elegans* [115], and rats [116]. Similarly, we also observed that ROS generation was directly linked to the exposure dose. The harmful effects of smaller particles (4 µm) were more prominent as compared to those induced by larger ones (10 and 20 µm). Oxidative stress caused by exposure to MNPLs is known to trigger a cascade of genotoxic events [49]. However, and in spite of their assumed risk of causing DNA damage, we still have many knowledge gaps in their genotoxicity to in vivo models. As DNA damage has been conclusively established to result in serious health consequences in humans [117], it is imperative that we gather as much data as possible on the genotoxic potential of MNPL exposure. In our study, we used a comet assay in hemocytes to assess whether PSNPLs could cause DNA strand breaks, finding significant increases in the extent of DNA damage following exposure to PSNPLs. Even though such effects were not dose-dependent, they varied according to particle size, with an indirect correlation between the level of damage and microparticle size. Recent reports involving tests on *Drosophila* seem to support the genotoxic impact of PSNPLs, including somatic mutation and recombination in a wing spot assay [45]. Furthermore, a more recent work on human testing of the impact of plastic particles on lung epithelial A549 cells found no genotoxic effects of pristine plastics, although the biodegraded MPLs caused damage to DNA [108]. This finding supports our findings on the genotoxicity of MNPLs in *Drosophila* once they were degraded and eroded inside the gut lumen of the larvae, with smaller particles (4 µm) causing greater DNA damage, especially at the highest dose. We should note that such results correspond to MPLs’ capacity to cause oxidative stress, whose correlation with genotoxicity and mutagenicity is well established [118,119]. Overall, our results show that exposure to PSMPLs via ingestion has genotoxic potential, as particularly smaller particles can end up in the hemolymph and induce significant DNA damage, depending on the dose of plastic material. We know that various types of pollutants may coexist in the same location and interact with each other, which raises more concern over the potential role of MNPLs as carriers for other more hazardous materials such as heavy metals, because the surface features of MNPLs enhance their chemical adsorption on toxic compounds even at low doses [8,120], or organic contaminants known to cause a range of diseases in mammals [114,121]. The MNPL samples from plastic debris in the North Atlantic Gyre have been detected to contain dangerous amounts of cadmium, titanium, and lead [122], and trace amounts of nickel, arsenic, and titanium [123]. Despite such huge health risks, there is still rather limited research to examine the possible effects of combined exposure to MPLs and heavy metal particles. Previous work in the field concluded that MNPLs might function as vectors to transport hazardous trace elements such as metals and organic pollutants [124]. Therefore, recent reports underline the need for further studies into the interaction between plastic and metals [120]. Most importantly, metal pollution has become prevalent in waters, soil, and air all around the world [125], and MPLs in the same environments inevitably combine with metal particles to create MNPLs+metal complexes. Such more hazardous particles can then escape into a variety of environments, inevitably coming into contact with food webs in the ecosystem, and yet very little research has so far covered their potential impact on terrestrial and aquatic life [8,55]. Furthermore, when compared to exposure to CdCl_2_ or AgNO_3_ alone, combined exposure to PSMPLs and CdCl_2_ or AgNO_3_ was found to result in more aggravated toxicity, as revealed by increased gut damage, oxidative stress, and DNA damage. The co-exposure of both plastics and metals could have induced additive or synergistic effects in *Drosophila* larvae, probably through PSMPLs transporting CdCl_2_ or AgNO_3_ into deeper tissues and releasing metal particles there, thus, causing increased toxic effects [39]. Likewise, a recent study on zebrafish has reported that combined exposure to PSMPLs and copper might be causing greater synergistic effects, with significant changes in malondialdehyde (MDA) and superoxide dismutase (SOD) in the liver and intestine of the model organism [40]. Considering all the results obtained from the current study, biological interactions (uptake, gut damage, oxidative stress, and DNA damage) between ingested PSMPLs and heavy metals (cadmium or silver) were confirmed in *Drosophila* larvae exposed by the feeding route.

## 5. Conclusions

*Drosophila* serves as a powerful testing model for the in vivo monitoring of the biodistribution of PSMPLs throughout the organ systems after their ingestion. A series of images clearly illustrating their presence in the gut lumen and their interactions with other lumen compartments, along with the hemolymph and hemocytes, provide compelling visual evidence. Despite egg-to-adult viability tests showing no significant toxicity of PSMPLs, we managed to collect pieces of molecular evidence that they induced significant oxidative stress and intestinal damage, along with DNA damage in hemocytes. The combined exposure to PSMPLs and heavy metals (cadmium or silver) induced marked adverse and genotoxic effects including oxidant damage, as evidenced by enhanced ROS generation and genetic damage. Such deleterious effects were mainly attributed to the size-dependent activities of PSMPLs, with smaller sized particles exerting greater damage.

## Figures and Tables

**Figure 1 biology-11-01470-f001:**
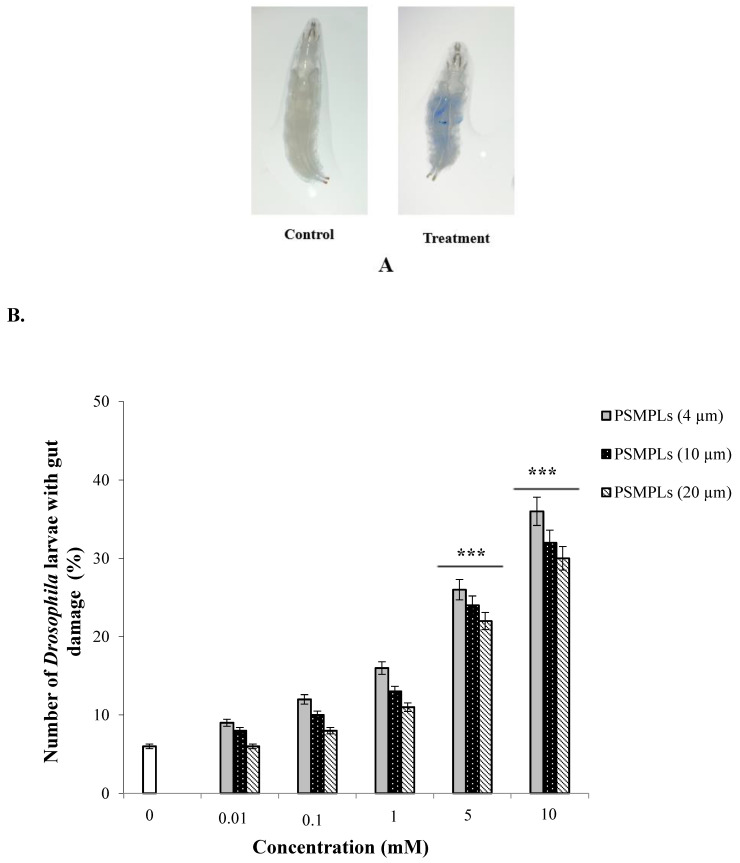
Intestinal damage in third-instar larvae after exposure to PSMPLs, CdCl_2,_ AgNO_3_, PSMPLs + CdCl_2_, and PSMPLs + AgNO_3_. (**A**) Trypan blue staining of damaged tissue was only observed in treated larvae (presence of gut damage). No tissue damage was observed in the control larvae. Percentage of gut damage in larvae after exposure to PSMPLs (**B**), CdCl_2_ and AgNO_3_ (**C**), PSMPLs + CdCl_2_ (**D**), and PSMPLs + AgNO_3_ (**E**). Data were analyzed by Mann–Whitney *U*-test. *** *p* ≤ 0.001, as compared to controls.

**Figure 2 biology-11-01470-f002:**
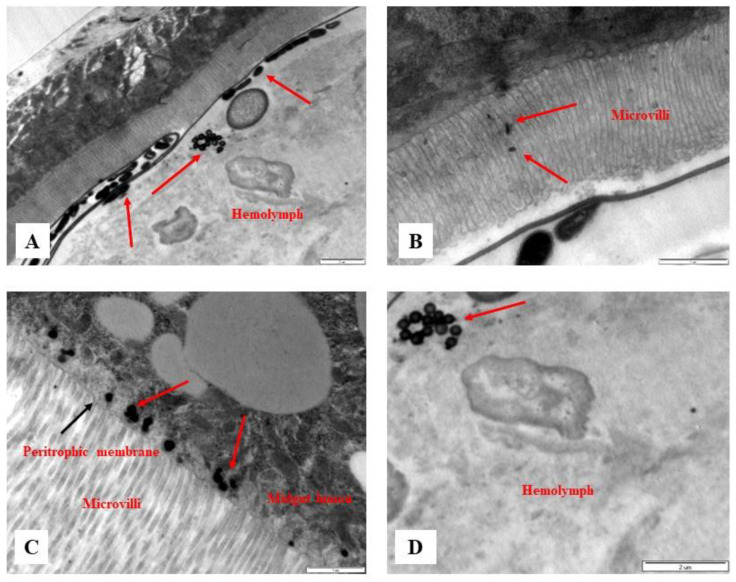
Images of ingested PSMPLs (4 µm) from their ingestion until their translocation to the hemolymph. PSMPLs (4 µm) are attached to the peritrophic membrane (**A**), inside the microvilli (**B**), distributed into the midgut lumen (**C**), and ultimately, reaching the hemolymph (**D**).

**Figure 3 biology-11-01470-f003:**
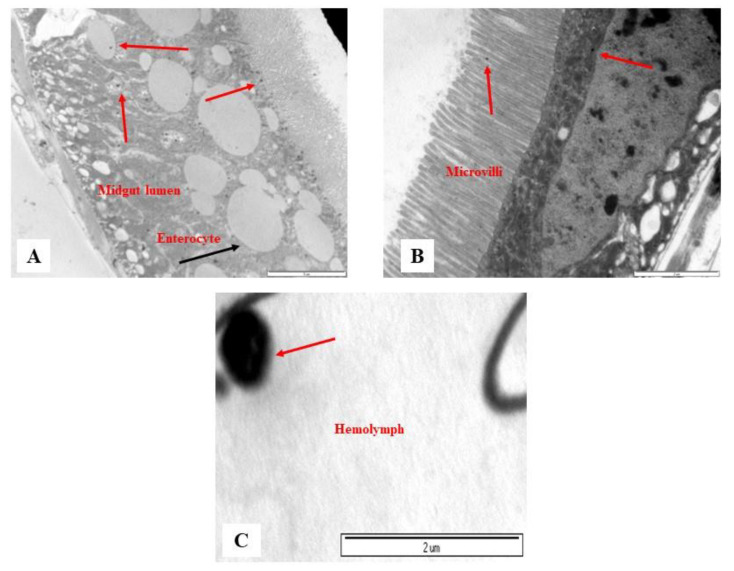
Images of ingested PSMPLs (10 and 20 µm) from their ingestion until their translocation to the hemolymph compartment. PSMPLs (10 µm) are distributed inside the midgut lumen (**A**), PSMPLs (20 µm) are inside the microvilli and distributed inside the midgut lumen (**B**) and, finally, reaching the hemolymph (**C**).

**Figure 4 biology-11-01470-f004:**
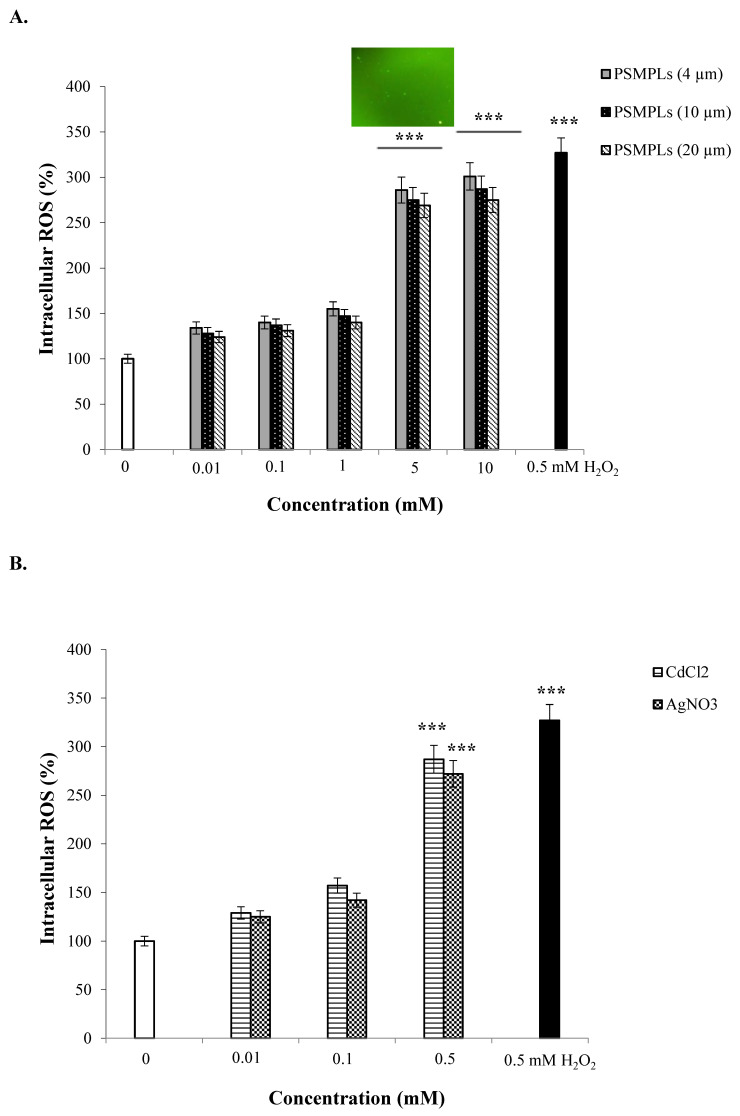
ROS production in hemocytes of third instar of untreated (0) and treated larvae exposed to different concentrations of PSMPLs (**A**), CdCl_2_ and AgNO_3_ (**B**), PSMPLs + CdCl_2_ (**C**), and PSMPLs + AgNO_3_ (**D**). Hemocytes were incubated with 5 μM of DCFH-DA at 24 °C for 30 min and observed using fluorescent microscopy. The fluorescence intensity of the hemocytes of exposed larvae with permethrin were quantified by ImageJ analysis. A total of 0.5 mM of H_2_O_2_ was used as positive control. *** *p* ≤ 0.001 when compared to the negative control by the Mann–Whitney U-test.

**Figure 5 biology-11-01470-f005:**
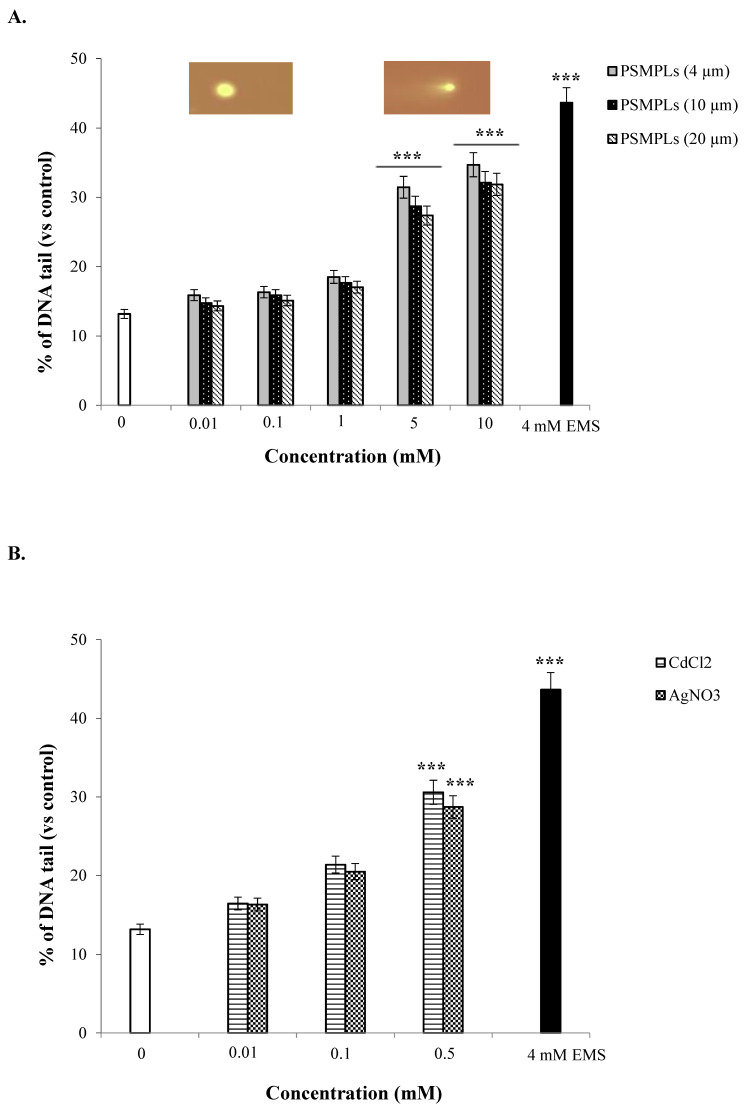
Genotoxic effects of PSMPLs (**A**), CdCl_2_ and AgNO_3_ (**B**), PSMPLs + CdCl_2_ (**C**), and PSMPLs + AgNO_3_ (**D**) in the Comet assay. Results indicate the % of DNA tail induced after the larvae were exposed to different doses of compounds for 24 h (three replicates were performed and 100 randomly selected cells were analyzed per treatment). Data represent the mean ± standard error (SE). EMS (4 mM) was used as positive control. *** *p* ≤ 0.001, as compared to the negative control by the Student’s *t*-test.

## Data Availability

All data generated or analyzed during this study are included in this published article.

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
