# Peer review of "Interactions of Ingested Polystyrene Microplastics with Heavy Metals (Cadmium or Silver) as Environmental Pollutants: A Comprehensive In Vivo Study Using Drosophila melanogaster"

_biology, 2022, doi:10.3390/biology11101470_

Round 1
Reviewer 1 Report
given in att

Author Response
Prof. Dr. Jukka Finne
Prof. Dr. Andrés Moya
Editor-in-Chief
Biology
Oliver Sun
Editor
Biology
Antalya, 21st September 2022
Manuscript ID: biology-1924143
Title: Interactions of Ingested Polystyrene Microplastics with Heavy Metals
(Cadmium or Silver) as Environmental Pollutants: A Comprehensive In Vivo
Study Using Drosophila melanogaster
Dear Prof. Dr. Jukka Finne,
Dear Prof. Dr. Andrés Moya,
Dear Oliver Sun,
Thank you very much for your kind letter, dated on 21st September, with the comments of the two reviewers that consider our work of interest, as well as suggestions of the revisions to this manuscript.
We have read carefully the comments of the reviewers and we expect that our responses and revisions will satisfy their suggestions. We hope that, with the revisions made in line with the comments/questions raised by the reviewers, the current version of our paper will be acceptable for publication.
In addition, we have introduced several modifications in the new version to ensure that the manuscript conforms to the format of the Journal. With the new version, I enclosed a marked with track changes copy where the modifications are highlighted.
I look forward to hearing from you at your earliest convenience.
Sincerely yours,
Prof. Dr. EÅŸref Demir
RESPONSE TO THE REVIEWERS
Reviewer: 1
The paper represents a study of synergistic effects of ingested polystyrene microplastics with
cadmium and/or silver as environmental pollutants using Drosophila melanogaster
as a model. The study is valuable from the aspect of different biological impact of various
pollutants which are increasing in the environment.
RESPONSE
We thank the Reviewer for the positive feedback.
In a comprehensive introduction the authors give review of literature data regarding
the background the research. However, that part should be more structured, less
broadly revue of environmental significance, but focused on the goal of their study.
The first part goes into wide details and review on microplastic, recycling,
biodegradation, global problems, etc. I am sure it can be halved.
RESPONSE
Thank you very much for your valuable contribution. Since microplastics and nanoplastics (MNPLs) pollution has a great importance in recent years, relevant information has been added to the introduction section so that readers can access detailed information about the effects, permanence and biological effects of MNPLs in the environment. In order to emphasize the importance of this study, it is aimed to draw attention to the transformations of MNPLs in the environment and the problems they cause.
- At the end of introduction, but not separated even with paragraph, the aim of the
study is given. It is a bit confusing and not fully consistent with the experimental
flow which proceeds in Material and Methods and Results. It should clearly state
the goals. Maybe the whole that part needs English revision to make it better, clear
and convincing. For example..“Therefore(?).this study aimed”first sentence is
ok,…than the methodology for one part (aim?) is given in details which is
inappropriate at that place. Last sentence even more weakens the consistency of
study design and goals: …”Furthermore (?), we also examined in great detail
(?)..the whole ..”..and it is actually the big part of the study..should not be given as
“also” etc…It is contradictory with the flow given in the results.
RESPONSE
Thank you very much for your valuable suggestion. We have revised the related sentences in line with your comments.
Material and Methods part is well covered with technical and methodology
information but I think that except in part 2.2. in the rest of the experiments
(staining, indigestion etc. the sample size.. the number of larvae examined, used,
presented in the results..is missing.
RESPONSE
We thank for this reflection. We have added the relevant information. On the other hand, carbon powder is not a stain. It is just used for better visibility of Drosophila eggs and added to Drosophila medium.
- Results. 3.1. Toxicity of PSMPLs and heavy metals (CdCl2 and AgNO3) Although
in Material and methods the part 2.2 is, as far as I understand, aimed to choose the
appropriate concentration and doses of Cd, Ag and PSMPLs administered
individually and simultaneously and to measure the effects through egg-to adult
survival, the presentation of the obtained results is missing important part. The
Figure 1 gives toxicity results measured as loss of viability (egg-to-adult survival),
but only for PSMPLs, CdCl2 and AgNO3 individually. There is no presented data on
combined effect which should be the main point of the study. The authors state that
“according to these results, PSMPLs “could be said” (to descriptive and speculative)
to have no statistically significant toxic effects as in D. melanogaster larvae”. And
that “significant toxic effects were observed after exposure to CdCl2 and AgNO3 at
doses higher than 0.5 mM “. In the whole study the combined, synergistic effects is
given and discussed except in this part. If they do not have these data than there the toxicity part is not completed and the authors should not consider it in this paper, just
keep shortly the methodological explanation of chosen doses. Even if there are
combined data (somehow missing in the results) my suggestion would be to skip this
part in the presentation form in the Results section and only explain which doses are
chosen.
RESPONSE
Thank you very much for your valuable contribution and different point of view. Viability experiments were previously carried out in our laboratory to determine the range of doses to be used in the genotoxicity studies. The criteria to choose the final selected doses were based on two reasons: First, a reduction in the percentage of developing treated larvae is a clear indication that the compounds affected the larvae and, in addition, the number of emerging adults must be high enough to perform the genotoxicity experiments. For this reason, we did not aim to find LC50 values in this study. Instead, we conducted a toxicity study to detect the genotoxic effects at doses above 50 percent viability.
The final sentence and emphasized in Conclusion should be deleted, or reephrased
and replaces in introduction, discussion…: “In conclusion, this study confirms that
Drosophila is an useful organism in environmental toxicity/geno-toxicity studies for
environmental pollutants.” Because that is not the main conclusion of this study “to
confirm” something well known from decades of genotoxicity studies on
Drosophila, including environmental pollutants such as heavy metals.
RESPONSE
Thank you very much for your valuable contribution and different point of view. We have deleted the relevant sentence.
- Reference list is too large... 146 references is too much for research arrticle, many
overlap and some fields are overrepresented.
RESPONSE
Thank you very much for your valuable contribution. A few references were deleted.

Reviewer 2 Report
Manuscript entitled “Interactions of Ingested Polystyrene Microplastics with Heavy Metals (Cadmium or Silver) as Environmental Pollutants: A Comprehensive In Vivo Study Using Drosophila melanogaster” by Demir et al. A very interesting study highlighting the toxic potential of microplastic with heavy metals. Overall data presented in the study showing the synergistic/additive toxic effect of microplastic with heavy metals. However, I have some major queries that is
1. How microplastics with cadmium and silver nitrate are environmentally relevant ? Any reports where combinations of microplastic with cadmium and silver nitrate are reported. If any please discuss and cite. Please discuss how exposed concentrations of cadmium and silver nitrate were selected and are these concentrations reported with microplastic.
2. In the Result section, Authors show larval viability in sterile water is 100% and after exposure to 10 mM to PSMPLs (4, 10, and 20 µm), 72, 75, and 77% of eggs successfully reached adult stage. Which is 28, 25 and 23% decrease in viability but authors show it is statistically nonsignificant. Please recheck statistics.
3. Is co-exposure of microplastic with Cadmium and silver nitrate to increase the toxic effects synergistically or additive? If the effect is synergistic or additive, I am still not sure whether microplastic interact with heavy metals as the title of the manuscript says.
4. Data presented in bar graph if authors present the data in bar graph with dot plot that clearly show the number of replicates and range
Author Response
Prof. Dr. Jukka Finne
Prof. Dr. Andrés Moya
Editor-in-Chief
Biology
Oliver Sun
Editor
Biology
Antalya, 21st September 2022
Manuscript ID: biology-1924143
Title: Interactions of Ingested Polystyrene Microplastics with Heavy Metals
(Cadmium or Silver) as Environmental Pollutants: A Comprehensive In Vivo
Study Using Drosophila melanogaster
Dear Prof. Dr. Jukka Finne,
Dear Prof. Dr. Andrés Moya,
Dear Oliver Sun,
Thank you very much for your kind letter, dated on 21st September, with the comments of the two reviewers that consider our work of interest, as well as suggestions of the revisions to this manuscript.
We have read carefully the comments of the reviewers and we expect that our responses and revisions will satisfy their suggestions. We hope that, with the revisions made in line with the comments/questions raised by the reviewers, the current version of our paper will be acceptable for publication.
In addition, we have introduced several modifications in the new version to ensure that the manuscript conforms to the format of the Journal. With the new version, I enclosed a marked with track changes copy where the modifications are highlighted.
I look forward to hearing from you at your earliest convenience.
Sincerely yours,
Prof. Dr. EÅŸref Demir
RESPONSE TO THE REVIEWERS
Reviewer: 2
Manuscript entitled “Interactions of Ingested Polystyrene Microplastics with Heavy Metals (Cadmium or Silver) as Environmental Pollutants: A Comprehensive In Vivo Study Using Drosophila melanogaster” by Demir et al. A very interesting study highlighting the toxic potential of microplastic with heavy metals. Overall data presented in the study showing the synergistic/additive toxic effect of microplastic with heavy metals. However, I have some major queries that is
RESPONSE
We thank the Reviewer for the positive feedback.
- How microplastics with cadmium and silver nitrate are environmentally relevant ? Any reports where combinations of microplastic with cadmium and silver nitrate are reported. If any please discuss and cite. Please discuss how exposed concentrations of cadmium and silver nitrate were selected and are these concentrations reported with microplastic.
RESPONSE
Thank you very much for your valuable contribution and different point of view.
Silver (Ag) content in the surface soils varies between <0.01 and 5 mg/kg (Sterckeman et al., 2002; Murray Rogers, and Kaufman, 2004). In addition, general concentration of cadmium (Cd) ranges from 0.1 to 1 mg/kg in soil (Alloway, 2013).
Microplastic (MPL) concentrations in rivers and inland waters now exceed alarming levels— samples from the river Elbe in Europe were shown to contain 3.35 x 106 microplastic particles in sediments and 5.57 MPLs per cubic meter (Scherer et al. 2020).
The concentrations of PSMPLs were based on our previous toxicity studies into particulate matter on Drosophila (0.01-10 mM) (Zhang et al. 2020; Demir 2021; Matthews et al. 2021). On the other hand, the concentrations of CdCl2 and AgNO3 were determined based on the data from previous toxicity and genotoxicity research with CdCl2 (0.01-0.2 mM) and AgNO3 (0.05, 0.2 and 1 mM) on Drosophila (Alaraby et al. 2022b; Zhang et al. 2020; Demir et al. 2011).
The studies in the literature and the amounts found in the environment were taken into consideration for choosing the tested concentrations.
- In the Result section, Authors show larval viability in sterile water is 100% and after exposure to 10 mM to PSMPLs (4, 10, and 20 µm), 72, 75, and 77% of eggs successfully reached adult stage. Which is 28, 25 and 23% decrease in viability but authors show it is statistically nonsignificant. Please recheck statistics.
RESPONSE
Thank you very much for your valuable contribution. For PSMPLs, our results showed that a partial decrease in viability was detected as dose-dependent. These reductions did not show statistically significance.
- Is co-exposure of microplastic with Cadmium and silver nitrate to increase the toxic effects synergistically or additive? If the effect is synergistic or additive, I am still not sure whether microplastic interact with heavy metals as the title of the manuscript says.
RESPONSE
We thank for this reflection. The results obtained in co-exposure of microplastic with Cadmium and silver nitrate are higher than in single exposures. In this context, it is considered that there is an interaction between microplastics and metals.
- Data presented in bar graph if authors present the data in bar graph with dot plot that clearly show the number of replicates and range
RESPONSE
Thank you very much for your valuable contribution and different point of view. We have revised the related graph in line with your comments.

Round 2
Reviewer 1 Report
The authors responded in 48h hours to serious comments I made which is impressive but obviously not enough.
1. Introduction is still not structured and is maybe appropriate for review paper, but for research article should be more straightforward.
2. RESULTS, Figure 1.
The Figure 1 gives toxicity results measured as loss of viability (egg-to-adult survival), but only for PSMPLs, CdCl2 and AgNO3 individually. There is no presented data on combined effect which should be the main point of the study. In the whole study the combined, synergistic effects is given and discussed except in this part. If they do not have these data than there the toxicity part is not completed and the authors should not consider it in this paper, just keep shortly the methodological explanation of chosen doses. Even if there are combined data (somehow missing in the results) my suggestion would be to skip this part in the presentation form in the Results section and only explain which doses are chosen. and describe the effect on viability. Without Figures.
Author Response
Prof. Dr. Jukka Finne
Prof. Dr. Andrés Moya
Editor-in-Chief
Biology
Oliver Sun
Editor
Biology
Antalya, 26th September 2022
Manuscript ID: biology-1924143
Title: Interactions of Ingested Polystyrene Microplastics with Heavy Metals
(Cadmium or Silver) as Environmental Pollutants: A Comprehensive In Vivo
Study Using Drosophila melanogaster
Dear Prof. Dr. Jukka Finne,
Dear Prof. Dr. Andrés Moya,
Dear Oliver Sun,
Thank you very much for your kind letter, dated on 26th September, with the comments of the two reviewers that consider our work of interest, as well as suggestions of the revisions to this manuscript.
We have read carefully the comments of the reviewers and we expect that our responses and revisions will satisfy their suggestions. We hope that, with the revisions made in line with the comments/questions raised by the reviewers, the current version of our paper will be acceptable for publication.
In addition, we have introduced several modifications in the new version to ensure that the manuscript conforms to the format of the Journal. With the new version, I enclosed a marked with track changes copy where the modifications are highlighted.
I look forward to hearing from you at your earliest convenience.
Sincerely yours,
Prof. Dr. EÅŸref Demir
RESPONSE TO THE REVIEWERS
Reviewer: 1
The authors responded in 48h hours to serious comments I made which is impressive but obviously not enough.
RESPONSE
We thank the Reviewer for the positive feedback.
- Introduction is still not structured and is maybe appropriate for review paper, but for research article should be more straightforward.
RESPONSE
Thank you very much for your valuable suggestion. We have revised the Introduction in line with your comments.
- RESULTS, Figure 1.
The Figure 1 gives toxicity results measured as loss of viability (egg-to-adult survival), but only for PSMPLs, CdCl2 and AgNO3 individually. There is no presented data on combined effect which should be the main point of the study. In the whole study the combined, synergistic effects is given and discussed except in this part. If they do not have these data than there the toxicity part is not completed and the authors should not consider it in this paper, just keep shortly the methodological explanation of chosen doses. Even if there are combined data (somehow missing in the results) my suggestion would be to skip this part in the presentation form in the Results section and only explain which doses are chosen. and describe the effect on viability. Without Figures.
RESPONSE
We thank for this reflection. We have introduced the suggestions in the manuscript. The Figure 1 was deleted. We only explained which doses are chosen and described the effect on viability.
Reviewer 2 Report
Authors satisfactory answered the queries raised by reviewer. Manuscript recommended for publication.
Author Response
Prof. Dr. Jukka Finne
Prof. Dr. Andrés Moya
Editor-in-Chief
Biology
Oliver Sun
Editor
Biology
Antalya, 26th September 2022
Manuscript ID: biology-1924143
Title: Interactions of Ingested Polystyrene Microplastics with Heavy Metals
(Cadmium or Silver) as Environmental Pollutants: A Comprehensive In Vivo
Study Using Drosophila melanogaster
Dear Prof. Dr. Jukka Finne,
Dear Prof. Dr. Andrés Moya,
Dear Oliver Sun,
Thank you very much for your kind letter, dated on 26th September, with the comments of the two reviewers that consider our work of interest, as well as suggestions of the revisions to this manuscript.
We have read carefully the comments of the reviewers and we expect that our responses and revisions will satisfy their suggestions. We hope that, with the revisions made in line with the comments/questions raised by the reviewers, the current version of our paper will be acceptable for publication.
In addition, we have introduced several modifications in the new version to ensure that the manuscript conforms to the format of the Journal. With the new version, I enclosed a marked with track changes copy where the modifications are highlighted.
I look forward to hearing from you at your earliest convenience.
Sincerely yours,
Prof. Dr. EÅŸref Demir
RESPONSE TO THE REVIEWERS
Reviewer: 2
Authors satisfactory answered the queries raised by reviewer. Manuscript recommended for publication.
RESPONSE
We thank the Reviewer for the positive feedback. Thank you very much for your valuable contribution and different point of view.